# Sports Value Consciousness Moderates the Effect of Exercise Behavior on Sport Activity Loyalty: A Moderated Mediation Model with Sports YouTube Engagement

**DOI:** 10.3390/bs13070583

**Published:** 2023-07-13

**Authors:** Youngtaek Oh

**Affiliations:** Department of Kinesiology & Sport Management, Texas A&M University, College Station, TX 77843, USA; oyt0712@tamu.edu; Tel.: +1-979-307-8471

**Keywords:** exercise behavior, sport activity loyalty, sports YouTube engagement, sports value consciousness, university student

## Abstract

This study aims to address a lack of research on sports value consciousness among Korean university students by examining the moderated mediating effect of sports value consciousness on the relationship between university students’ sports YouTube engagement and their sport activity loyalty. A total of 231 Korean university students were recruited for this study. The analytical model was validated using the SPSS PROCESS Macro (Model no. 8). Exercise behavior was found to have a significant positive effect on both sports YouTube engagement and sport activity loyalty, while sports YouTube engagement had a significant positive effect on sport activity loyalty. Furthermore, the interaction between exercise behavior and sports value consciousness had a significant positive effect on sport activity loyalty at a low to moderate level. Results suggest that, for Korean university students, sports value consciousness is more important at a low to moderate level of sport activity loyalty than at a high level. The importance of sports value consciousness is highlighted in the context of the development of sports media and the sociological aspects of sports engagement among university students.

## 1. Introduction

How much do Korean university students value sports? Most people pursue a healthy and happy lifestyle through exercise behavior [1]. Depending on how much they value sports, individuals can realize self-management [2], health [3], well-being [4], and happiness [5] goals. Various researchers have reported that the rise of sports media, such as YouTube sports programs, has positively influenced exercise behavior [6] and exercise intention [7]. Hanani [8] study highlights the sports activity value of recreational participants in the field of sports social science. However, research on the sports value consciousness of university students not majoring in kinesiology is lacking.

Sports value consciousness refers to awareness of the values and principles related to sports and physical activity. These values include fair play, respect for opponents, teamwork, and dedication [9]. Sports value consciousness involves recognizing the broader social and ethical significance of sports and physical activity and applying those principles to the improvement of society [10,11]. Previous studies have reported significant effects of sports value consciousness on sport activity loyalty [12], exercise self-efficacy, and quality of life [13]. Studies related to sports value consciousness have mainly been conducted with elite athletes [4] and kinesiology major students [14,15]. This study is the first research effort on the level of sports value consciousness and sports media engagement among non-kinesiology major university students, filling a gap noted by Kavetsos et al. [16], who emphasized the need for research on sports value analysis, claiming that it has not been sufficiently investigated.

This study explores the relationship between sports value consciousness and sports YouTube engagement, which has been influenced by the development of sports media. Ye [17] explained that the rapid development and dissemination of network information technology has had a positive effect on the sports consciousness and sports behavior of university students. By examining the sociological and psychological aspects of sports through sports media, this study aims to contribute to the advancement of sports social science. This research could help not only Korean students but also students from other countries understand the importance of sports and sports value consciousness from a diplomatic perspective through understanding the development of sports media and sports social science. Universities can use the results of this study both to strengthen physical education-related courses with the goal of enhancing sports value consciousness and to develop strategies to take advantage of sports media and sport activity loyalty. Therefore, the purpose of this study is to examine the relationship between sports YouTube engagement and sport activity loyalty among Korean university students, based on the level of interaction between exercise behavior and sports value consciousness.

### 1.1. Literature Review

#### 1.1.1. The Relationship between Exercise Behaviors, Sports YouTube Engagement, and Sport Activity Loyalty

Korean university students have the option to take physical education courses as part of their general education requirements; however, students who do not value physical activity tend to opt for theory-based or general education courses. In this section, we will examine the relationship between exercise behaviors and engagement in sports YouTube content, as well as sport activity loyalty. Regular and sufficient physical activity is known to improve health [18]. Haskell et al. [19] explain that moderate-intensity aerobic physical activity or high-intensity aerobic exercise (e.g., inline skating or running) for at least 30 min a day prevents health issues such as heart failure, diabetes mellitus, hypertension, and specific types of cancer. According to the European Commission [20], 40% of European citizens participate in physical activity at least once a week, with 7% (>30 million) participating in physical activity five or more times a week, and 42% (>72 million) engaging in high-intensity physical activity. Sports play an important role in the economy of European nations. In Korea, the 7330 campaign emphasizes the importance of participating in physical activity for at least 30 min three times a week. While changing sedentary habits can be challenging [21,22], previous research has shown that regular physical activity has a positive impact on subjective well-being [23].

In this study, the Godin Leisure-Time Exercise Questionnaire scale developed by Godin et al. [24] was used to assess the exercise behavior of college students. Participants were asked to report the amount of time they spent engaging in exercise over the course of one week. The score was calculated by multiplying the number of high-intensity bouts by nine, moderate-intensity bouts by five, and low-intensity bouts by three, and then summing the total score to determine the level of exercise behavior. The first hypothesis of this study is related to the relationship between exercise behaviors and sport activity loyalty. Sport activity loyalty can be influenced by various factors such as personal enjoyment of the activity, social relationships, and perceived health benefits of the activity [25]. Additionally, loyalty to a specific sports team or organization can be developed and maintained through victories and defeats [26]. Most previous studies have examined the relevance of sports teams [26], sports brands [27], and sports events [28]. This study explores the relationship between sport activity loyalty and exercise behaviors, which has been underexplored in the sport social science literature.

The next hypothesis to be tested involves the relationship between exercise behavior and sports YouTube engagement. YouTube is a website for user-generated content (UGC), similar to Flickr, Facebook, and Wikipedia [29]. It offers a much better user experience and attracts more users than similar online services such as Vimeo [30]. The YouTube interface allows users to quickly access videos and switch from one clip to another. YouTube creators can create user account pages and upload content on topics of interest. Subscribers can leave comments and receive notifications when new content is uploaded, which provides convenience to users [29]. As of 2022, YouTube has over 2.1 billion global users and is the most popular social media platform among American adults [31]. In 2021, 81% of American adults reported using YouTube, and this usage rate continues to steadily increase, surpassing other social media platforms such as Facebook (61%), Instagram (40%), and Pinterest (31%) [31]. YouTube provides a variety of sports-related information to the public [32]. Previous research has demonstrated relationships between interest in sports participation and fitness participation motivation [33], exercise practice [34], sports lessons [35], and doing bodies [36] through YouTube engagement. In addition to social science research, studies in sports natural science fields such as sports taping [37] and rehabilitation [38] are also being conducted. We expect YouTube engagement to improve exercise behavior among non-kinesiology major university students.

The third hypothesis is related to the relationship between YouTube engagement and sport activity loyalty. Sports YouTube engagement has been reported to have significant associations with sports fan brand awareness [39], brand image and purchase intention [40], and sports fan loyalty [41]. Most of these studies were conducted from a sports management perspective; this study aims to examine the social aspects of sports in relation to YouTube engagement. The three research hypotheses are based on theory and previous studies as follows.

**Hypothesis** **1.**
*Exercise behaviors will have a positive effect on sport activity loyalty.*


**Hypothesis** **2.**
*Exercise behaviors will have a positive effect on sports YouTube engagement.*


**Hypothesis** **3.**
*Sports YouTube engagement will have a positive effect on sport activity loyalty.*


#### 1.1.2. Relationship between Sports Value Consciousness and Variables

Sports value consciousness refers to various values, beliefs, attitudes, and behaviors related to sports [42]. Sports value consciousness can be formed through various experiences [43] for example, individual preferences or cultural values around sports can influence sports value consciousness. Enjoying and actively participating in sports in individual or societal contexts can shape sports value consciousness. If physical activity experiences have a positive effect on personal growth and development, participants may develop values related to sports [4]. Personal sports experiences can contribute to forming social bonds and empathy as they are shared within society [44]. Because sports play a role in social integration, health promotion, and cultural exchange, sports value consciousness also has implications for society [45], students’ behavior [46], happiness [47], and subjective well-being [4]. Hypotheses 4 and 5 address the relationship between sports value consciousness and exercise behavior among university students.

**Hypothesis** **4.**
*The effect of interaction between exercise behavior and sports value consciousness on sports YouTube engagement.*


**Hypothesis** **5.**
*The effect of interaction between exercise behavior and sports value consciousness on sport activity loyalty.*


## 2. Method

### 2.1. Participants

The participants in this study are university students in Korea in 2023. The survey was conducted online during the spring semester of 2023, targeting 231 students currently enrolled in university. The general characteristics of the participants are as follows: 144 males (62.3%) and 87 females (37.7%), with 71 first-year students (30.7%), 70 second-year students (30.3%), 58 third-year students (25.1%), and 32 fourth-year students (13.9%).

### 2.2. Measures

#### Questionnaire Scales

The questionnaire comprised scales of reliability and validity which had been adequately assessed in previous studies. First, to check exercise behaviors, the Godin et al. [24] scale was used to measure the level of exercise behavior, including high, medium, and low intensity. Specifically, the total exercise participation time measurement was multiplied by 9 points for high intensity, 5 points for medium intensity, and 3 points for low intensity, and the weekly exercise behavior time score was calculated as follows: (9 × high intensity) + (5 × medium intensity) + (3 × low intensity). To measure sports YouTube engagement, the Rubin [48] scale was supplemented and used and consisted of 17 questions. Participants responded to each question on a 7-point Likert-type scale. The Cronbach alpha for this scale was 0.95, indicating high internal consistency. The results of confirmatory factor analysis (CFA) indicated a good fit for the data (*χ*^2^ = 258.215, d*f* = 89, *p* < 0.001, Q = 2.901, IFI = 0.946, TLI = 0.927, CFI = 0.946, RMSEA = 0.091). Additionally, the composite reliability (CR) was 0.96, indicating high reliability of the measure, and the average variance extraction (AVE) was 0.56.

To measure sport activity loyalty, the Yang [49] scale was used. The scale consisted of 10 items and was measured on a 5-point Likert-type scale. The Cronbach alpha for this scale was 0.92, indicating high internal consistency. The results of the CFA showed a slightly high value for the RMSEA (*χ*^2^ = 16.816, d*f* = 4, *p* < 0.001, Q = 4.204, IFI = 0.988, TLI = 0.956, CFI = 0.988, RMSEA = 0.118, CR = 0.86, AVE = 0.50). Additionally, the CR was 0.86, indicating high reliability of the measure, and the AVE was 0.50. Next, to measure sports values consciousness, the Spreitzer et al. [50] scale was used. The scale consisted of 15 items and was measured on a 5-point Likert-type scale. The Cronbach alpha for this scale was 0.88, indicating high internal consistency. The results of the CFA showed a good fit for the data (*χ*^2^ = 53.584, d*f* = 19, *p* < 0.001, Q = 2.820, IFI = 0.962, TLI = 0.943, CFI = 0.962, RMSEA = 0.089). Additionally, the CR was 0.91, indicating high reliability of the measure, and the AVE was 0.58. The data were used after informed consent from the study subjects.

### 2.3. Statistical Analysis

The collected data were analyzed using the SPSS 24.0 (SPSS Inc., Chicago, IL, USA), SPSS PROCESS Macro 2.13, and Amos 24.0 (IBM, New York, NY, USA) statistical programs. First, a frequency analysis was conducted. Second, Cronbach’s alpha values were calculated to check the reliability of each measurement tool, and CFAs were performed to find out the validity of the constructs. Third, Pearson’s product-moment correlation was calculated on major variables. Fourth, SPSS PROCESS Macro (Model 8; Hayes, [51]) was used to test the moderated mediation effect of sports value consciousness on the relationship between exercise behavior and sports YouTube engagement and sport activity loyalty. Finally, the conditional indirect effect was tested.

## 3. Results

### 3.1. Result of Statistical and Correlation Analyses

The descriptive statistics of study variables, including mean, standard deviation, skewness, and kurtosis, are listed in Table 1. Correlations were performed to examine overall relationships between variables, and all variables were found to be correlated below 0.66 (see Table 1; [52]).

### 3.2. Path Analysis of Dimensions of Exercise Behavior, Sports YouTube Engagement, and Sport Activity Loyalty

All the direct effects among each variable are shown in Figure 1 and Table 2, which presents the relationship among exercise behavior, sports YouTube engagement, and sport activity loyalty. Exercise behavior had a significant positive effect on sport activity loyalty (B = 0.002, t = 2.185, *p* < 0.05), Hypothesis 1 was accepted. Exercise behavior had a significant positive effect on the sports YouTube engagement (B = 0.003, t = 4.239, *p* < 0.001), Hypothesis 2 was accepted. Finally, sports YouTube engagement had a significant positive effect on sport activity loyalty (B = 0.527, t = 6.356, *p* < 0.001), Hypothesis 3 was accepted.

### 3.3. Moderated Mediating Effects of Sports YouTube Engagement in the Relationship between Exercise Behavior and Sport Activity Loyalty According to Sports Value Consciousness

The interaction between exercise behavior and sports value consciousness had a not significant effect on the sports YouTube engagement (B = 0.001, t = 0.460, *p* > 0.05), Hypothesis 4 was rejected. In addition, the interaction between exercise behavior and sports value consciousness had a significant effect on sport activity loyalty (B = −0.003, t = −2.962, *p* < 0.01). After specifically checking the interaction value, medium (95% CI = 0.001 to 0.004), and low levels were found to be meaningful (95% CI = 0.002 to 0.006), Hypothesis 5 was accepted (Figure 2). That is, the interaction between exercise behavior and sports value consciousness showed a significant effect on sport activity loyalty only at low and medium levels.

## 4. Discussion

This study was designed from a sociological perspective to examine the relationship between sports value consciousness of non-kinesiology major university students and exercise behavior, sports media development, and sport activity loyalty. Participation in exercise behavior positively influenced sport activity loyalty. Results demonstrated no relationship between college students’ perceived sports value consciousness and sports YouTube engagement, but sports value consciousness was shown to have a significant (low to moderate) impact on sport activity loyalty. Non-kinesiology major university students with a moderate level of sports value consciousness demonstrated higher sports loyalty than those with a high level of sports value consciousness. Based on the research results, the following discussion is presented.

The first hypothesis of this study is related to the relationship between exercise behaviors and sport activity loyalty. Sport activity loyalty can be influenced by various factors such as personal enjoyment of the activity, social relationships, and perceived health benefits of the activity [25]. Additionally, loyalty to a specific sports team or organization can be developed and maintained through victories and defeats [26]. Most previous studies have examined the relevance of sports teams [26], sports brands [27], and sports events [28]. This study explores the relationship between sport activity loyalty and exercise behaviors, which has been underexplored in the sport social science literature.

The result of this study is supported by García-Fernández et al. [53], who showed that participating in fitness activities positively affected sport activity loyalty in a sample of 1805 fitness participants, and by Bodet [54], who found that participating in fitness activities positively affected sport loyalty in a sample of 252 fitness participants. Moreover, Funk et al. [26] found that individuals who participated in sports developed stronger emotional responses, functional knowledge, and greater symbolic value related to sports teams, indirectly supporting this study. It is widely known that most people associate sport activity loyalty with an affinity for physical activity. Although most studies on this subject have been conducted in the field of sports management, the present study finds support for these ideas from a social science and psychological perspective. Our results support the aim of encouraging exercise behavior to enhance attachment and pride in sports.

Exercise behavior had a significant impact on engagement in sports YouTube content, consistent with Sui et al. [55], who showed a sharp increase in the viewing of online home training and fitness videos, including popular videos hosted on free video streaming platforms such as YouTube and Instagram, pre and post-COVID-19. Our findings are also consistent with those of Kim et al. [56], who analyzed YouTube video titles during the COVID-19 pandemic and found that the term “exercise” had the highest frequency. Specifically, terms such as “hip-up” and “body profile” appeared most frequently. Clearly, college students interested in physical activity can easily access exercise-related information on YouTube. Our results demonstrate the utility of YouTube for acquiring exercise-related information—particularly important for those with a sports value consciousness and exercise behavior who demonstrate sport activity loyalty.

Sports YouTube engagement had a significant influence on sport activity loyalty, consistent with Checchinato et al. [57], who showed that the presence of sports brands is enhanced through the content produced by sports clubs (official content) and fans (UGC) on YouTube. Our results also align with those of Spencer [34], who emphasized the use of YouTube as a tool for spreading and learning Brazilian jiu-jitsu techniques. Sports YouTube programs have rapidly grown in popularity in recent years, receiving considerable attention due to their professional commentary and coverage of sports games from various perspectives [58]. YouTube is an efficient platform for users to increase their understanding of sports, prepare for new challenges, and gain new exercise experience. Sports YouTube engagement can play a crucial role in individuals’ lives, especially young adults like college students, by providing them with opportunities to enjoy sports and maintain a healthy lifestyle [32]. This study highlights the benefits of sports YouTube engagement.

The interaction between exercise behavior and sports value consciousness did not show a significant relationship with sports YouTube engagement, but it did have a significant impact on sport activity loyalty at a low to moderate level. These results are consistent with Oh et al. [12], who investigated Korean (*n* = 383) and Chinese (*n* = 1368) university students and found that the interaction between physical activity and sports value consciousness had a significant impact on sport activity loyalty, and with Sato et al. [59], who examined 328 ski tourists and found individuals who perceived the value of sports had increased sport activity loyalty. University students who engage in physical activity and have sports value consciousness tend to recognize the importance of sports; when they perceive the pleasure of sports, such as health benefits, enjoyment with friends and family, and self-development, they tend to increase their sport activity loyalty. University students who understand the value of sports are more likely to actively participate in sports and maintain their activity levels.

This study aimed to investigate the relationship between exercise behavior and sport activity loyalty among university students in Korea, based on their sports value consciousness. The findings provide fundamental data for understanding the culture of physical activity among Korean university students. In addition, the use of social media, particularly sports YouTube engagement, could be promoted as a positive antecedent to stimulate university students’ exercise behavior and sport activity loyalty. This study’s results also have practical implications for encouraging physical activity participation among less active female university students, the general population with little information on exercise, and older adults.

## 5. Conclusions

As a result of validating an SPSS PROCESS Macro model with 231 Korean university students, this study found that exercise behavior and sports YouTube engagement both positively influenced sport activity loyalty. The interaction between exercise behavior and sports value consciousness did not have a significant effect on sports YouTube engagement but had a significant effect on sport activity loyalty at a low to moderate level. These results suggest that, rather than focusing on high levels of sports value consciousness, teaching university students to recognize sport activity loyalty at an appropriate level through general physical education courses could help them understand sports value appropriately.

Limitations and future research directions suggested by this study center on research subject demographics. This study targeted non-kinesiology major university students in Korea; future research should include a comparative analysis of kinesiology major and non-major university students. Previous studies, such as Oh et al. [12], compared the sports value consciousness of Korean and Chinese university students. Comparative analyses of the sports value consciousness of students from Korea and other countries would be a useful avenue for future research. Finally, social media plays an important role as a variable in sports media and sports value consciousness and is an important area of study for sports sociology.

## Figures and Tables

**Figure 1 behavsci-13-00583-f001:**
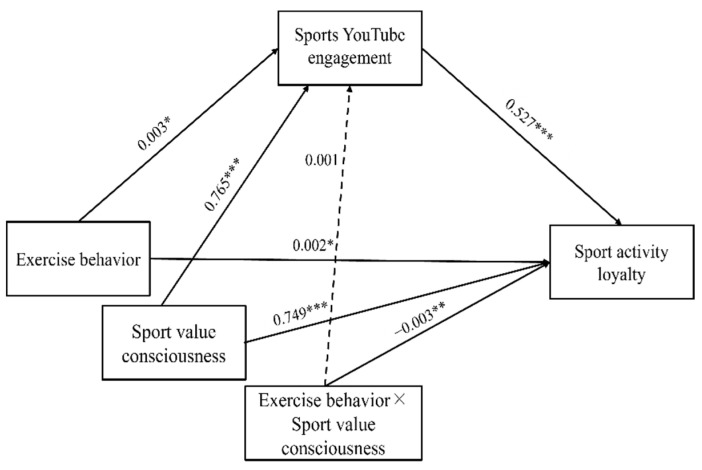
Finalized hypothesized model. Note. * *p* < 0.05, ** *p* < 0.01, *** *p* < 0.001.

**Figure 2 behavsci-13-00583-f002:**
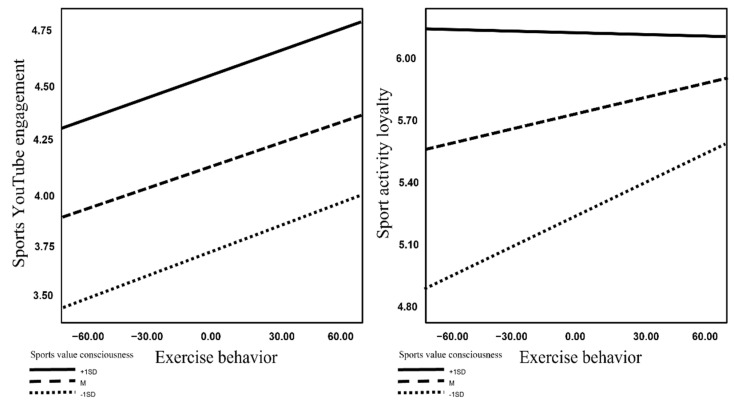
Interaction effects graph.

**Table 1 behavsci-13-00583-t001:** Pearson correlations of scores. Descriptive statistics and Cronbach’s α.

Variable	1	2	3	4
1. Exercise behavior	-	-	-	-
2. Sports YouTube engagement	0.16 *	-	-	-
3. Sport activity loyalty	0.15 *	0.66 **	-	-
4. Sports value consciousness	−0.09	0.62 **	0.66 **	-
Mean	74.51	4.10	5.71	4.29
Standard deviation	65.04	0.76	1.10	0.63
Skewness	1.99	−0.72	−1.12	−1.22
Kurtosis	5.57	0.37	1.89	2.77

Note. * *p* < *0*.05, ** *p* < 0.01.

**Table 2 behavsci-13-00583-t002:** Direct and Interaction effect index.

Variable	B	S.E	Boot S.E	t(*p*)	95% CI	
					LLCI	ULCI
Sports YouTube engagement: R^2^ = 0.424, F (3, 226) = 55.492, *p* < 0.001						
Constant	4.102	0.039		106.399 ***	4.026	4.178
Exercise behavior	0.003	0.001		4.239 ***	0.001	0.004
Sports value consciousness	0.765	0.062		12.437 ***	0.644	0.887
Exercise behavior × Sports value consciousness	0.001	0.001		0.460	−0.001	0.002
Sport activity loyalty: R^2^ = 0.573, F (4, 225) = 75.533, *p* < 0.001						
Constant	3.532	0.344		10.281 ***	2.855	4.210
Exercise behavior	0.002	0.001		2.185 *	0.001	0.003
Sports YouTube engagement	0.527	0.083		6.356 ***	0.364	0.691
Sports value consciousness	0.749	0.099		7.523 ***	0.553	0.946
Exercise behavior × Sports value consciousness	−0.003	0.001		−2.962 **	−0.006	−0.001
Conditional effects of the focal predictor at values of the moderator(s):					Boot LLCI	Boot ULCI
−1SD	0.004		0.001	4.057 ***	0.002	0.006
Mean	0.002		0.001	2.185 *	0.001	0.004
+1SD	−0.001		0.001	−0.317	−0.003	0.002

Note. LL, UL: bias-corrected 95% confidence interval (lower limit, upper limit); bias-corrected bootstrapping method is conducted for indirect effect estimates; * *p* < 0.05, ** *p* < 0.01, *** *p* < 0.001.

## Data Availability

The data presented in this study are available on request from the corresponding author. The data are not publicly available due to privacy issues.

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
