# Peer review of "Sports Value Consciousness Moderates the Effect of Exercise Behavior on Sport Activity Loyalty: A Moderated Mediation Model with Sports YouTube Engagement"

_behavsci, 2023, doi:10.3390/bs13070583_

Round 1

Reviewer 1 Report

In the abstract, the word "effect" is not suitable, in that the relationship between variables is correlation rather than casual effect. Please check.

Line 36-37, the concept of "Sports value consciousness refers to an individual's awareness of values and principles related to sports and physical activity." should be added reference.

In the last paragraph of introduction, it better to add the description of your research design briefly.

In the literature review section, the concepts of “exercise behaviors, sports YouTube engagement, and sport activity loyalty” should be interpreted.

In the "2.2.1. Questionnaire Scales", related references concerning the rationalities of employing the adopted questionnairs should be added.

Author Response

Response to Reviewer 1 Comments

Title: The moderating role of sports value consciousness on the relationship between exercise behavior and sport activity loyalty: A moderated mediation model incorporating sports YouTube

 The title has been modified to clarify the research topic.

Title: The moderating role of sports value consciousness on the rela-tionship between exercise behavior and sport activity loyalty: A moderated mediation model incorporating sports YouTube engagement

Point 1: In the abstract, the word"effect" is not suitable , in that the relationship between variables is correlation rather than casual effect. Please check.

Response 1: Thank you for your feedback.

The abstract addresses the influence of the independent variable on the mediator and dependent variables, specifically emphasizing the absence of correlation. This refers to the pathway of the regression analysis. The areas that need minor corrections have been marked in red. Thank you.

Point 2: Line36-37, the concept of "Sports value consciousness refers to an individual's awareness of values and princi- 37 ples related to sports and physical activity." should be added reference.

Response 2: Thank you for your feedback. The references have been provided.

Point 3: in the last paragraph of introduction, it better to add the discription of your research design briefly.

Response 3: Thank you for your feedback. We made an effort to present the purpose of the study clearly. The modified portion is highlighted in red.

Therefore, the purpose of this study is to examine the relationship between sports YouTube engagement and sport activity loyalty among Korean university students, based on the level of interaction between exercise behavior and sports value consciousness.

Thank you for your positive review.

 I believe that receiving feedback has enhanced the overall quality of your thesis. I appreciate your positive feedback.

Reviewer 2 Report

The paper entitled The Moderating Role of Sports Value Consciousness on the 2 Relationship between Exercise Behavior and Sport Activity 3 Loyalty: A Moderated Mediation Model Incorporating Sports 4 YouTube, aims to analyze aims to examine the moderated mediating effect of sports value consciousness on the relationship between university students sports YouTube engagement and their sport activity loyalty. The paper is well written and has a great and strong methodology. This study employs a robustness analysis. However, the introduction and discussion are not persuasive enough that the findings make a significant contribution to the literature and could therefore override these limitations. I include some comments below related to this summary for consideration.

1.     In relation to the contribution of the study to the literature, I did not get a sense from the article that the findings revealed anything other than what we already know. Please clarified that;

2.     The introduction of the paper was very descriptive, it did not situate the current study in literature or highlight what the gap in the literature is that this study is trying to address. At least, the authors should situate better the main purposes of this study;

3.     The discussion is very descriptive and any statements about the contribution and conclusions of the study are not new. At least this moment. Please clarified better and justified your choices.

4.     Overall, the paper has conditions for being accepted in Behavioral Sciences, however, the authors should clarify the points above.

No comments.

Author Response

Point 1: In relation to the contribution of the study to the literature, I did not get a sense from the article that the findings revealed anything other than what we already know. Please clarified that;

Point 2: The introduction of the paper was very descriptive, it did not situate the current study in literature or highlight what the gap in the literature is that this study is trying to address. At least, the authors should situate better the main purposes of this study;

Response 1, 2: Thank you for your feedback. In order to clearly articulate the purpose of the study, it is presented in the final paragraph of the introduction. This reflects the effort made to explain the study's focus. Thank you.

Additionally, English experts have corrected the awkward expressions in the sentence, aiming to emphasize the necessity and purpose of a clear thesis as a research document through descriptive writing.

Point 3: The discussion is very descriptive and any statements about the contribution and conclusions of the study are not new. At least this moment. Please clarified better and justified your choices.

Response 3:

The descriptive expressions in the thesis writing format have been modified extensively. Moreover, it is essential for this study to avoid fragmentary conclusions in the dissertation.

This study possesses the following characteristics:

The purpose of this study is to examine the relationship between sports YouTube engagement, sport activity loyalty, sports behavior, and sports value consciousness among college students. The researcher obtained results indicating that subjects exhibit higher sports activity loyalty when their exercise behavior and sports value consciousness levels are low.

This study has implications for the systematic planning and progression of future research endeavors. It aims to compare sports value consciousness among college students majoring in kinesiology and non-kinesiology, focusing on whether non-kinesiology students show greater awareness of sports activity loyalty when perceiving a low level of sports participation and sports value consciousness. Therefore, this study highlights the importance of continuous research and provides direction for future investigations.

I believe that the editors will fully grasp the researcher's position and make a positive decision based on it.

Round 2

Reviewer 1 Report

In your abstract and paper, there are many “effect”, while your research is a cross sectional study but not longitudinal one. So the word "effect" is not suitable, and I suggest you change it into correlation or predictor. Please check.

Reviewer 2 Report

No more comments. The paper meet the conditions to be accept. 

Author Response

We appreciate the positive feedback received from the reviewers and thank them for their valuable input, which has enhanced the quality of the paper.

Round 3

Reviewer 1 Report

I am sorry that the difference between "effect" and "correlation" has not be fully clarified. It is not a longitudinal study, so "effect" is not suitable here, which will confuse readers.